# A Spatial Analysis of the Prevalence of Female Genital Mutilation/Cutting among 0–14-Year-Old Girls in Kenya

**DOI:** 10.3390/ijerph16214155

**Published:** 2019-10-28

**Authors:** Ngianga-Bakwin Kandala, Chibuzor Christopher Nnanatu, Glory Atilola, Paul Komba, Lubanzadio Mavatikua, Zhuzhi Moore, Gerry Mackie, Bettina Shell-Duncan

**Affiliations:** 1Department of Mathematics, Physics & Electrical Engineering (MPEE), Northumbria University, Newcastle NE 18 ST, UK; chibuzor.nnanatu@northumbria.ac.uk (C.C.N.); glory.atilola@northumbria.ac.uk (G.A.); paul.komba@northumbria.ac.uk (P.K.); lubanzadio.mavatikua@northumbria.ac.uk (L.M.); 2Independent Consultant, Vienna, Virginia, VA 22182 USA; zhuzhimoore@gmail.com; 3Department of Political Science, University of California, San Diego, CA 92093-0521, USA; gmackie@ucsd.edu; 4Department of Anthropology, University of Washington, Seattle, WA 98195-3100, USA; bsd@u.washington.edu

**Keywords:** FGM/C, spatial modelling and mapping, social norms, space-time interactions

## Abstract

Female genital mutilation/cutting (FGM/C), also known as female circumcision, is a global public health and human rights problem affecting women and girls. Several concerted efforts to eliminate the practice are underway in several sub-Saharan African countries where the practice is most prevalent. Studies have reported variations in the practice with some countries experiencing relatively slow decline in prevalence. This study investigates the roles of normative influences and related risk factors (e.g., geographic location) on the persistence of FGM/C among 0–14 years old girls in Kenya. The key objective is to identify and map hotspots (high risk regions). We fitted spatial and spatio-temporal models in a Bayesian hierarchical regression framework on two datasets extracted from successive Kenya Demographic and Health Surveys (KDHS) from 1998 to 2014. The models were implemented in R statistical software using Markov Chain Monte Carlo (MCMC) techniques for parameters estimation, while model fit and assessment employed deviance information criterion (DIC) and effective sample size (ESS). Results showed that daughters of cut women were highly likely to be cut. Also, the likelihood of a girl being cut increased with the proportion of women in the community (1) who were cut (2) who supported FGM/C continuation, and (3) who believed FGM/C was a religious obligation. Other key risk factors included living in the northeastern region; belonging to the Kisii or Somali ethnic groups and being of Muslim background. These findings offered a clearer picture of the dynamics of FGM/C in Kenya and will aid targeted interventions through bespoke policymaking and implementations.

## 1. Introduction

Female genital mutilation/cutting (FGM/C) also known as female circumcision is defined as a procedure involving the removal of all or part of the external female genitalia for non-medical reasons. It is estimated that 200 million girls and women from 30 countries mainly in Africa, the Middle East and Asia have experienced the practice [1]; and approximately 3 million young girls are at risk of being cut each year [2]. FGM/C is a topical issue under the United Nations Sustainable Development Goals (SDGs) which seek to achieve the elimination of all harmful practices against women and girls by the year 2030. The FGM/C-elimination goal is premised on the idea that FGM/C is a gender-based violence which violates the rights of women and girls and represents a huge public health concern. There are both short and long-term health consequences associated with FGM/C. Shock, bleeding and severe pains are among the short-term consequences [3,4,5,6]. While in the long term, the individual may have trouble passing urine, lack sexual libido, have increased risk of childbirth complications and other psychological issues [7,8,9].

In line with the SDGs year 2030 target, international and national bodies as well as government and non-governmental organisations have deployed several intervention programmes aimed at the abandonment of the practice in most of the high prevalent countries. However, the desired pace of change has been slow [10,11,12]. The study by Shell-Duncan et al. [13] showed evidence of variation in the prevalence of the practice with respect to some characteristics. In the context of Kenya, which has experienced national FGM/C prevalence drops from 38% in 1998 to 21% in 2014 [13], research has shown that prevalence of FGM/C was high throughout among Kisii and Somali women and almost zero among younger women (15–19 years) of Kalenjin, Kamba, Kikuyu and Taita/Taveta ethnic groups [13]. Another study by Achia [14] concluded that female genital mutilation/cutting among 15–49 years old women in Kenya was deeply rooted in culture and religion and that there was a significant correlation existing between a woman’s current FGM/C status and her support for the continuation of the practice. In essence, Achia found a higher proportion of cut women supporting the continuation of FGM/C. Similarly, in the context of Nigeria, Kandala et al. [15] found that a woman’s support for the continuation of FGM/C and her FGM/C status were key predictors of her daughter’s likelihood of experiencing FGM/C.

### The Theory behind Normative Influences on FGM/C Prevalence

Among the theories developed to account for FGM/C, and to understand variation in the pace of FGM/C elimination, the social convention/norm theory appears to have influenced FGM/C elimination strategies. However, it has also been the least tested factors in the quantitative FGM/C research, especially in the context of Kenya. In essence, social norms theory posits that patterns of behaviour are influenced by unofficial social rules that are learned through social interactions with salient people in the community, commonly referred to as a reference group [16]. These are held in place by reciprocal expectations of those in the reference group. As a result, actions by an individual are not driven solely by their own preferences and attitudes. Rather they are also influenced by perceived expectations of others and pressure to conform. In a model first developed by a political scientist, Gerald Mackie, FGM/C was hypothesised to be a social norm that spread and became locked in place by interdependent expectations regarding marriageability [17]. In competition for marrying higher social class, FGM/C provided an advantage by signalling fidelity, and became a universal prerequisite for marriage. On the other hand, work by Shell-Duncan and colleagues on Senegambia suggested that the practice serves as a signal to other circumcised women that a circumcised girl or woman has been trained to respect the authority of her circumcised elders and is worthy of inclusion in their social network [18]. FGM/C became locked in place, since those opting out would pay the high price of foregoing marriage, legitimate childbearing and ostracization.

In addition to social norms are *moral norms*, which rest on internalized values of right and wrong, and *legal norms*, formally codified in law [19]. *Religious norms* may be experienced as moral norms when they result in internalized beliefs about behaviour, or as social norms when they are upheld by beliefs that a behaviour is required by a religious community. A sub-type of social norms called *gender norms* are defined as what are appropriate behaviours for upholding culturally constructed ideals on masculinity or femininity by Cislaghi & Heise [19]. Discriminatory gender norms limit girls’ and women’s access to power within their families and communities, reduce their educational and economic opportunities, and alter their own aspiration and ambitions for their lives.

Normative influences as used in our context, therefore, include the combination of the norms defined above assessed by a woman’s FGM/C status/the proportion of cut women within a given community (social norms), the proportion of women in the community who supported the continuation of the practice (moral norms), and the proportion of women in the community who believed that FGM/C was a religious obligation (religious norms).

The findings from Achia, 2014, Shell-Duncan et. al, 2017 and Kandala et. al., 2019 [13,14,15] prompt two key questions: Do we find similar trends among the younger generation of Kenyan girls 0–14 years old, and are there similar strong associations between normative influences and a girl’s FGM/C status? Another crucial question is whether the geographical location of an individual really matters in this context, and if so, by what magnitude did these unobserved effects of geographical (spatial) locations influence the observed FGM/C prevalence? The present study examined how social norms influenced spatial variations in the risk of FGM in girls 0–14 in Kenya. It sought to test the predictions of the social norms/social convention theory. In other words, we examined the trends of FGM/C prevalence among 0–14 girls in Kenya focusing on the influence of social norms. We then assessed the effects of an individual’s geographical location as well as of time on the observed prevalence.

The above aims were explored using advanced statistical approaches which allowed multiple adjustments of the key individual- and community-level factors, whilst controlling for unobserved spatial and temporal effects. This Bayesian hierarchical modelling approach popular in disease mapping studies [20,21], has been applied only sparingly in the case of FGM/C [14,22,23,24]. While a similar approach adopted here was used in the context of Nigeria in Kandala et.al, 2009; Kandala and Shell-Duncan, 2019 [22,24], the work in Achia, 2014 [14] employed Bayesian logistic regression models in the context of Kenya for the detection of spatial clustering of FGM/C among women aged 15–49 years in Kenya using only 2008 Kenya Demographic and Health Survey (KDHS) data. Our approach is novel and different from Achia, 2014 [14] in a number of ways: Firstly, this is the initial attempt to model and map FGM/C among Kenyan girls aged 0–14 years using Bayesian Hierarchical approach and focusing on the roles of normative influences. Secondly, while the study by Achia, 2014 [14] used only one survey data, in particular KDHS 2008, our approach utilised four successive KDHS surveys data for 1998, 2003, 2008 and 2014, making it possible to assess both spatial and temporal variations in the practice. Finally, using Markov chain Monte Carlo (MCMC) approach in Bayesian framework allowed seamless and robust parameters estimation in the face of complex sampling design used in the KDHS.

The remainder of this paper is structured as follows. The methods including data and variables descriptions are given in Section 2 and we give the rationale behind the adoption of our statistical analytical approach and outline the analytical procedures employed. The results obtained from the various analyses conducted in this study including descriptive and spatial analyses are presented in Section 3. In Section 4, we discuss the implications of our findings on the public health and wellbeing of women and girls, as suggested by our findings and outline the limitations of the study respectively. We conclude the paper in Section 5. Note that throughout the paper we used *cut* and *circumcised* interchangeably to refer to a woman or girl who experienced FGM/C.

## 2. Methods

### 2.1. Data

As already stated, datasets were extracted from four successive nationally representative Kenya Demographic and Health Surveys (KDHS) spanning across sixteen years period, namely, 1998 KDHS, 2003 KDHS, 2008 KDHS and 2014 KDHS. We drew on responses to the core household questionnaire as well as the module on FGM/C administered to women aged 15–49 years. The module on FGM/C includes three key sections: (1) whether the woman underwent FGM/C or not, and details about the event, (2) whether the respondent’s daughter(s) underwent FGM/C or not, and details about that event and (3) the woman’s opinion about the continuation of the practice. All women who received the full survey were asked if they had ever heard of “female circumcision”. Those who had not were asked if they had ever heard of a practice in which a girl has part of her genitals cut.

Across the four waves of surveys, different methods were used to obtain information on the FGM/C status of daughters. The 1998 and 2003 KDHS obtained information on the circumcision status of the eldest daughter. The 2008 KDHS asked women about the FGM/C status of their most recently cut daughter aged 0 to 14 years, and the 2014 KDHS asked women about the circumcision status of all daughters aged 0 to 14 years.

Each round of survey used a stratified, two-stage cluster sampling design. In the 2003 KDHS sample, urban areas were oversampled and sparsely populated and largely nomadic areas in the North Eastern Province contributed few numbers because of difficulties in reaching this population. Therefore, using the 1999 population census frame, a total of 400 clusters consisting of 129 urban and 271 rural locations, were selected from the master frame.

In 2008, 995 clusters in rural areas and 617 clusters in urban areas were randomly selected out of 5360 clusters in the Kenya Population and Housing Census. These clusters approximated villages and urban neighbourhoods [25]. From each selected cluster, 25 households were selected at random without replacement to make a representative sample of households at the national level of the eight provinces in Kenya.

In 2014, a total of 5360 clusters split into four equal subsamples were drawn with a stratified probability proportional to size from 96,251 enumeration areas (EAs) of the 2009 Kenya Population and Housing Census. At the end, the sample was designed to have 40,300 households from 1612 clusters nationally, with 995 clusters in rural areas and 617 in urban areas. The final samples of daughters’ records provided data on 4069, 4048, 7195 and 12,434 girls for 1998, 2003, 2008 and 2014 KDHS respectively.

For our purposes, we used only two datasets, namely, the most recent DHS survey data, 2014 KDHS, and pooled dataset of the KDHS from 2003 to 2014 to assess the roles of normative influences and examine trends in the practice over time. The reason for excluding KDHS 1998 in the pooled data was because there were no KDHS in the north-eastern province of Kenya in 1998.

### 2.2. Outcome Variable

Throughout this study we used only one outcome variable, a binary outcome, which indicated the FGM/C status of a girl. However, as already highlighted above, the outcome variable was defined differently in earlier surveys. In 1998 and 2003, the outcome variable was whether the eldest daughter (aged 0–14) of the woman was circumcised. In 2008, the outcome variable was the most recently circumcised daughter (aged 0–14), and in 2014 the outcome variable was whether any daughter (aged 0–14) was circumcised. The outcome variable was coded 1 if the girl was cut, otherwise coded 0 if she was uncut for each survey year.

### 2.3. Exposure Variables

The exposure variables for the study included individual- and community-level characteristics as well as the geographical location of a girl and her mother. A mother’s location (urban vs rural, region of residence) captures the fact that if FGM/C is a social norm, people who are more geographically proximate are more likely to interact, and therefore have a normative influence upon one another. Hence, geographical location as an exposure variable serves as a proxy measure of normative influences that cannot otherwise be measured (unobserved). These normative influences were assessed using individual-level (a woman’s FGM/C status) and community-level characteristics (proportion of cut women in the community, proportion of women in the community who support FGM/C continuation (not in 2003 KDHS), proportion of women in the community who believed FGM/C was required by religion (not in 2003 KDHS). Other characteristics assessed included socio-demographic variables such as age of a girl and her mother, mother’s education, ethnicity and household wealth. In addition, we derived a variable called ethnic fractionalization index (*EFI*). This is a community-level continuous variable with values ranging from 0 to 1, which measures the level of ethnic mixing of a given community. The *EFI* is calculated via a decreasing transformation of the Herfindahl concentration index [26] given as
(1)$$EFI=1-\sum_{k=1}^{n}s_{k}^{2}$$
where $s_{k}$ is the proportion of the $k^{th}$ ethnic group in a given community with n≥2 ethnic groups. A higher value (close to 1) of *EFI* indicates a multi-ethnic community in which all the available ethnic groups are fairly equal in size, while a lower value (close to zero) indicates a community with fewer ethnic mixing or monoethnic (if zero). The idea behind the inclusion of the *EFI* variable in the models is due to the hypothesis that different social norms may be influenced by different reference groups and one possibility is for change to align with groups that tolerate a desired change. In regions where there is little ethnic diversity, and where the prevalence of FGM/C is near-universal, there may be little opportunity to shift to a reference group that does not uphold the practice [27,28,29,30,31,32].

### 2.4. Statistical Analysis

#### 2.4.1. Bivariate Data Analysis

First, we carried out weighted bivariate analyses with Stata version 14 applying the *svy* command [33]; then we tested the pairwise strength of association between the covariates and the response variable. The weighted outputs were cross tabulated and reported (Table A1
Appendix A).

#### 2.4.2. Bayesian Hierarchical Spatial and Space-Time Modelling

Considering some analytical challenges posed by the use of cluster sampling (mainly due to inherently dependent observations) by KDHS in the selection of their respondents, there is need to employ a valid statistical approach that accounts for correlated responses whilst controlling for other linear and non-linear continuous covariates. Consequently, we used a class of mixed models called structural additive regression (STAR) models [34,35], to estimate the effects of different covariates on the observed data. Unlike the standard regression model, which assumes strictly linear relationship between the covariates and the response variable, these models allowed us to account for spatial autocorrelation and heterogeneity, inherent in such a geographically referenced data by extending the model to the so-called *geo-additive models* [36]. Thus, we fitted Bayesian hierarchical geo-additive regression models to the two datasets extracted from the surveys for our purposes, namely, the 2014 KDHS and the pooled 2003 to 2014 KDHS data. Using this advanced statistical technique, we controlled for spatial dependence of observations in neighbouring regions as well as spatial heterogeneity of observations from distant neighbours within Kenyan regions in a coherent Bayesian logistic regression framework. Following Nnanatu et al. (2019), we extended the modelling approach to allow for interactions in time and space for improved estimates and to obtain prevalence maps with finer resolution. Details of the model development are found in the Appendix section of Nnanatu et al., 2019 [23]. However, we briefly describe the model below.

We denote the outcome variable of whether girl i was cut or not by $y_{i}$ such that $y_{i}$ is one (1) or zero (0) depending on whether girl i was cut or uncut. The random variable $y_{i}$, therefore has a Bernoulli distribution with parameter $p_{i}$. Given the logit link function $\eta_{i}$, the fully adjusted model for the pooled data is given in (2) as
(2)$$\eta_{i}=logit\left(p_{i}\right)=\log\left(\frac{p_{i}}{1-p_{i}}\right)=\beta_{0}+z_{i}^{\prime }\beta +f_{1}\left(x_{i1}\right)+\ldots +f_{p}\left(x_{ip}\right)+f_{y}\left(year\right)+f_{str}\left(s_{i}\right)+f_{unstr}\left(s_{i}\right)+f_{st}\left(s,t\right),$$
where $p_{i}=p_{ist}$ is the probability of cutting girl i in region s at time $t;\;f_{1}\left(.\right),\ldots ,f_{p}\left(.\right)$ are the functions (no necessarily smooth) of non-linear continuous covariates, $x_{i}s$ such as age, time effects etc. $\beta_{0}$ is the intercept, $\beta ={\left(\beta_{1},\;\ldots ,\;\beta_{p}\right)}^{\prime }$ are unknown coefficients of other class of covariates, $z_{i}s$; $s_{i}=1,\ldots ,S,$ where $s_{i}$ is the geographically referenced location of girl i, $f_{str}\left(.\right)$ and $f_{unstr}\left(.\right)$ denote the structured (correlated) and the unstructured (uncorrelated) spatial effects, respectively; $f_{st}\left(s,t\right)$ is a non-linear smooth function of the unobserved interactions in space and time.

We employed full Bayesian inference framework via Markov Chain Mote Carlo (MCMC) techniques. For our purposes, in order to allow for spatial autocorrelation and borrow strength from neighbouring regions, we assigned Markov random fields prior [37].
(3)$$f_{str}\left(s\right)|f_{str}\left(r\right),\tau_{s};r\neq s∼N\left(\sum_{r∼s}\frac{f_{str}\left(r\right)}{N_{s}},\frac{\tau_{s}^{2}}{N_{s}}\right),$$

To the structured (correlated) component of spatial effects, $f_{str}\left(.\right).\;N_{s}$ denotes the number of adjacent regions to region s; $\tau_{s}>0$ is a smoothness parameter. Other possible priors include Gaussian random fields priors and the types examined by Lee (2011) which include convolution model, Cressie model (Cressie, 1993) and Leroux model (Leroux et al., 1999). However, we note that choice of Markov random fields (MRF) priors for the structured spatial effect was made following a rigorous sensitivity analysis using other priors supported by BayesX, such as Gaussian random fields (GRF) with MRF having an edge over other priors. For example, the dependence introduced via the MRF prior ensures that information is shared among neighbouring regions. In other words, MRF priors allows us to “borrow strength” from neighbours in order to cope with the sample variation of the region effect and obtain estimates for areas that may have inadequate sample sizes or areas that were not sampled (see, for example, Mollié, 1996). As a result, there is a reduced variability of estimates between sparsely populated regions and densely populated neighbours.

Finally, a computational advantage is gained from the sparseness introduced by the conditional independence structure of the MRF in that the full conditional distribution of each structured spatial component is easily computed with greatly minimized computational cost.

Note that r∼s implies that regions s and *r* are neighbours. Also, to account for spatial heterogeneity, we assigned independent and identically distributed zero mean Gaussian prior to the unstructured (uncorrelated) spatial component, $f_{unstr}\left(.\right)$,
(4)$$f_{unstr}\left(s\right)|\tau_{u}∼N\left(0,\;\tau_{u}^{2}\right),$$
where $\tau_{u}>0$ is a variance parameter.

We model the continuous variables assumed to be non-linear (smooth) functions using Bayesian P(enalised)-splines [38] which are the Bayesian analogue of the P-splines developed by Eilers and Marx [39], These continuous metrics include; time (year) variable ($f_{I}\left(year\right)$, mother’s age, ($f_{i}\left(agem\right)$, girl’s age ($f_{i}\left(ageg\right)$), space-time interaction term ($f_{st}\left(s,t\right))$, ethnic fractionalization index ($f_{i}\left(efi\right)$), proportion of cut women in the community ($f_{i}\left(cut\right)$), proportion of women in the community who believed that FGM/C was a religious requirement ($f_{i}\left(req\right)$) and proportion of women who supported FGM/C continuation $f_{i}\left(sup\right)$. The smooth function $f_{j}\left(.\right)$, is expressed as a linear combination of n B-spline basis functions as
(5)$$f_{j}\left(.\right)=\underset{k}{\sum }\beta_{jk}B_{jk}\left(.\right).$$

The interaction term function $f_{st}\left(st\right)$ is modelled using the tensor product of one-dimensional B-splines
(6)$$f_{st}\left(st\right)=\underset{k}{\sum }\underset{j}{\sum }\beta_{jk\;}B_{sk}\left(s\right)B_{tj}\left(t\right),$$

Furthermore, we assigned Inverse Gamma prior distribution, to the variance parameters, that is, $\tau_{l}∼IG\left(a_{l},\;b_{l}\right)$, where $a_{.}$ and $b_{.}$ are hyperparameters. Using an advanced statistical software, BayesX, implemented in R statistical programming software through its R interface, namely R2BayesX [40], the unknown parameters were estimated in a Markov Chain Monte Carlo (MCMC) framework [41]. For optimal performance of the MCMC, we set $a_{.}=1,\;b_{.}=0.0005$ and drew N (=$2\times 10^{4})$ samples from the parameters space $\theta =\left\{\beta_{0},\;\left\{\beta_{j}\right\},\;\{f_{str}\left(.\right)\right\},\;\left\{f_{unstr}\left(.\right)\right\},\;\left\{p_{i}\right\}\}$ as well as the hyperparameters space $\phi =\left\{\tau_{\beta },\;\tau_{s},\;\tau_{u}\right\}$. We used a combination of thinning and Burn-in to improve the posterior estimates. Specifically, every 50th sample was stored (thinned) after discarding the first *B* (=2 × 10^3^) iterations as burn-in.

Finally, we used deviance information criterion (DIC) [42] and effective sample size (ESS), for assessment of model fit and model complexity. For the DIC, the smaller the better while a larger ESS signals a better fit model. Results (posterior estimates) obtained from the remaining N−B samples of the best fit models are reported in the appropriate tables, graphs and maps.

## 3. Results

In Figure 1, we show the eight administrative regions of Kenya. Later, we show how estimates from our models indicated variations in prevalence of FGM/C among 0–14 years old girls within a given region and across regions.

### 3.1. Descriptive Analysis

The data contained a total of 27,746 girls from 1998 (N = 4069), 2003 (N = 4048), 2008 (N = 7195) and 2014 (N = 12,434). Results from the bivariate descriptive analyses relating to the normative influence surrogate variables are shown in Table 1. Although, national prevalence of FGM/C among 0–14 years old girls in Kenya has decreased over the years from 9.9% in 1998 to 3.0% in 2014, wide variations in prevalence existed at the sub-national level and within strata of characteristics of the mother. For example, FGM/C prevalence among 0–14 years girls whose mothers were cut remained around 22% from 1998 to 2008 and later dropped to 9.9% in 2014. Similarly, FGM/C prevalence among girls whose mothers supported the continuation of the practice or whose mothers believed that FGM/C was a religious obligation, remained between 23% and 47% across the survey years.

### 3.2. Regional Evolution of FGM/C Prevalence among 0–14 Years Old Girls in Kenya

Figure 2 shows the evolution of FGM/C prevalence among 0–14 years old girls in Kenya with respect to their regions of residence (geographical location) across the four survey years. In 1998, there were no surveys in the north-eastern region. Crude FGM/C prevalence was highest at around 23% in Nyanza region, while the least crude prevalence of around 0.2% to 4.8% were found in the Coast, Nairobi, Central and Western regions. In Rift Valley and Eastern regions, FGM/C crude prevalence stood between 9% and 14%. The Kenya Demographic and Health Survey of 2003 included the north-eastern region and the highest crude FGM/C prevalence of around 61% was found in the north-eastern region. Also, prevalence in the Coast region slightly increased to around 12%, while reductions in prevalence were observed in Nairobi, Central and Eastern regions in 2003. In 2008, a slight reduction in prevalence was observed in the Rift Valley with a much larger reduction observed in the Coast. Prevalence remained high around 66% in the north-eastern region in 2008 with no clear changes observed in the other regions. Figure 2 further shows an overall drastic drop in FGM/C prevalence in 2014.

While the majority of the regions including Coast, Eastern, Central, Nairobi, Rift Valley and Western have seen a drastic drop in their rates, crude prevalence in the North-eastern and Nyanza remained relatively high (close to 42% and 11%, respectively). We note that Figure 2 shows huge crude spatial variations in the practice without controlling for other potential confounders that might explain the observed differences. While interpreting the observed spatial patterns of FGM/C across Kenyan regions as shown in Figure 2, it is important to bear in mind the different approaches adopted in the various surveys for measuring the FGM/C variable. However, we have utilized a sophisticated approach to account for potential errors due to sampling complexities and the possible effects of other unmeasured covariates. These are modelled as random effects thus minimizing uncertainties in parameters estimation.

### 3.3. Bayesian Hierarchical Geo-Additive Logistic Regression

The descriptions of the models fitted to the datasets along with the corresponding fit indices are given in Table 2. Three different models were fitted to each of the datasets. In relation to the 2014 KDHS, we first fitted the unadjusted model or Model A which contains only the normative influence “surrogate” variables. These variables included mother’s FGM/C status, proportion of cut women, proportion of women who supported the continuation of the practice, as well as the proportion of women who believed FGM/C was a religious obligation. Secondly, we fitted the space-adjusted model or Model B which accounted for the unobserved effects of spatial locations and normative influences simultaneously, with no recourse to other potential confounders such as religion and ethnicity. Thirdly, we deploy the so-called fully adjusted model or Model C to account for other potential confounders as well as unobserved spatial effects. A similar modelling strategy was adopted for the pooled dataset within the coherent Bayesian hierarchical logistic regression model approach. Model I, Model II and Model III were fitted to the pooled data. While Model I and Model II were equivalent to Model A and Model B fitted for the 2014 KDHS, Model III has a temporal effect as well as space-time interaction in addition to total spatial effects and other potential confounders.

Preliminary analysis using Moran’s I test [43] showed evidence of spatial autocorrelation in the data. Results obtained from the models described above are presented in Table A2 and Table A3 for 2014 KDHS data and the pooled 2003 to 2014 data, respectively.

### 3.4. 2014 KDHS

Across the three models fitted to the 2014 KDHS, normative influences were found to be key drivers of FGM/C prevalence among 0–14 years old girls in Kenya. The unadjusted model (Model A) showed that a girl whose mother was cut was about 3.94 times (95%CI = 2.05, 7.63) more likely to be cut than her counterparts born to uncut mothers. After adjusting for other potential confounders, the likelihood of cutting a girl whose mother was circumcised remained high, but no longer significant with a posterior odds ratio, POR = 1.97 (95%CI = 0.69, 6.01).

Figure 3 shows the non-linear effects of the community-level surrogate variables for the normative influences across the three models. Figure 3a was based on the unadjusted model (Model A) which did not account for the unobserved influence of space as well as the effects of other potential confounders. On the other hand, Figure 3b,c were obtained from Model B and Model C, respectively. We compared the behaviour of the three variables in each of the graphs. For the unadjusted model, there was no clear effect of proportion of women who supported FGM/C continuation on the likelihood of their daughters experiencing FGM/C. However, the unadjusted model showed that a girl’s likelihood of being cut increased with the proportion of women in the community who believed FGM/C was a religious obligation, and also increased rapidly when the proportion of cut women in the community exceeded 50% (Figure 3a).

After adjusting for other potential confounders, the likelihood of a girl being cut was found to increase with the proportions of women in her community who were cut, supported FGM/C continuation, believed FGM/C was a religious obligation (Figure 3a,b). Note that across the three models there was a sharp increase in the likelihood of girls being cut when the proportion of cut women within a girl’s community exceeded 60% and up to 80%.

Figure 4 shows the estimated unobserved total spatial effects of a girl’s geographic location on her likelihood of experiencing FGM/C in Kenyan regions. The left panel contains the spatial effects maps for Model B while the spatial effect maps for the fully adjusted model (Model C) are shown on the right panel of Figure 4. The mean posterior spatial effects maps (a) together with the 95% posterior significance maps (b) obtained from Model B fitted to the 2014 KDHS data indicate that there was significantly high risk of FGM/C due to unobserved geographical location effects for a girl who lived in Nyanza region. However, after controlling for other potential confounders (Figure 4c,d), this became nonsignificant along with other regions except Rift Valley region which remained a significantly low risk region in both cases. This implies that the likelihood of experiencing FGM/C for a girl who lived in Rift Valley was significantly unimpacted by her geographical location.

Other factors considered in the fully adjusted model (Model C) included some socio-demographic variables such as age, religion, marital status, ethnicity, mother’s occupation, mother’s highest level of educational attainment, type of place of residence (urban or rural) and ethnic fractionalization index (*EFI*). Results in Table A2 under Model C showed that type of place of residence (urban-region), had no significant effect on a girl’s likelihood of being cut. A girl who professed Muslim faith was about 5.5 times more likely to be cut than her Christian counterparts (95%CI = 2.65, 10.60). In addition, a girl who lived in a lower wealth quintile household had higher likelihood (POR = 1.21, 95%CI = 0.76, 1.99) of experiencing FGM/C than another girl who lived in a Middle wealth quintile household. However, this is not significant. Furthermore, a significantly high risk of cutting was found among girls who belong to Kisii ethnic group (POR = 11.73, 95%CI = 3.69, 37.38).

A girl whose mother had no education had higher likelihood of being cut than her counterpart whose mother obtained higher education. It is surprising to see that a girl whose mother had primary (POR = 0.71, 95%CI = 0.19, 2.27) or secondary (POR = 0.76, 95%CI = 0.23, 2.46) was less likely to be cut than girls with better educated mothers, but these were not statistically significant. In sum, based upon 2014 KDHS data, FGM/C was performed on Kenyan girls aged 0–14 years regardless of whether their mothers were well educated or not educated at all.

In terms of mother’s occupation, girls whose mothers never worked had lower likelihood of experiencing FGM/C than girls whose mothers had formal occupation. The fully adjusted model further showed that daughters of women who were in polygamous unions and women whose husbands beat, had higher likelihood of experiencing FGM/C than their counterparts with dissimilar situations. Other prominent factors as identified by Model C included a woman’s decision making power and marital age difference. It was found that daughters of women whose husbands or partners had sole decision-making power relating to her health and large household purchases, had higher likelihood of undergoing FGM/C than their counterparts whose mothers had the sole decision-making powers on such issues. Similarly, women who were older than their husbands were more likely to have their daughters cut than their counterparts.

Non-linear effects of a girl and her mother’s age as well as ethnic fractionalization index obtained from the fully adjusted model were shown in Figure 5.

Figure 5a shows that for the 2014 KDHS, the likelihood of a girl’s cutting decreased as her mother’s age increased. This inverse relationship found between mother’s age and her daughter’s likelihood of being circumcised was in contrast with the graph obtained for the effects of a girl’s age (Figure 5b). It can be observed from Figure 5b than a girl’s likelihood of experiencing FGM/C increased with her age and even more rapidly for girls who were aged 3–10 years old. A rather interesting result was found in Figure 5c where it was found that a girl’s likelihood of cutting decreased when the ethnic fractionalization index exceeded 0.4. This result indicated that the presence of higher number of different ethnic groups in a fairly equal proportion within a given community, minimises the spate of adherence to the practice of FGM/C within the community.

### 3.5. Pooled 2003 to 2014 Data

Table A3 shows the results obtained from the Bayesian Hierarchical spatial and spatio-temporal models fitted to the pooled 2003 KDHS to 2014 KDHS data. Across the models, normative influence surrogate variables predominated as key drivers of FGM/C among 0–14 years old girls in Kenya. Daughters of cut women had higher chances of being cut than their counterparts, with POR = 24.22 (95%CI = 16.62, 37.35) for Model I, POR = 25.40 (95%CI = 17.00, 33.57) for Model II, and POR = 27.29 (95%CI = 24.62, 30.38) for Model III. In addition, with respect to Model III, living in an urban region (POR = 1.17, 95%CI = 0.92, 1.51) has no statistically significant effect on a girl’s likelihood of being cut. Also, there is a significant effect of religion on a girl’s likelihood of being cut. Being a Muslim (POR = 2.48, 95%CI = 1.77, 3.35) makes a girl more likely to be cut than her counterparts. Belonging to Kisii (POR = 12.07, 95%CI = 5.79, 27.4), Somali (POR = 4.39, 95%CI = 2.12, 9.79), Maasai (POR = 4.17, 95%CI = 1.83, 9.45), Taita-taveta (POR = 3.59, 95%CI = 1.72, 8.91), Luhya and Other (POR = 2.17, 95%CI = 1.07, 4.76) ethnic groups significantly increased a girl’s likelihood of experiencing FGM/C.

Girls born to mothers with no education faced the highest risk of being cut (POR = 5.78, 95%CI = 3.27, 10.71). That probability decreased as the educational attainment level of mothers increased. There was no clear effect of mother’s occupation on the likelihood of her daughter experiencing FGM/C. No significant difference existed between the likelihood of a girl from a Middle wealth quintile household and that of another girl from lower or lowest wealth quintile household. However, girls who belonged to higher or highest wealth quintile household had smaller likelihood of experiencing FGM/C. Currently married women had higher likelihood of having their daughters cut than the never married and formerly married women.

Figure 6 shows the spatial effects mean posterior maps with the corresponding 95% posterior probability maps obtained after adjusting for the unobserved effects of geographical locations only (left panel), and after adjusting for unobserved location effects and other potential confounders including temporal and space-time interactions effects (right panel). The mean posterior maps for Model II together with the corresponding 95% posterior probability maps indicates that Nyanza and North-eastern regions were significantly high-risk regions while Central region was significantly low FGM/C risk region. After adjusting for potential confounders in Model III, it was found in Figure 6c,d that eastern and North-eastern regions were significantly high FGM/C risk region largely occasioned by unobserved location-specific effects.

We present, in Figure 7, the predicted fully adjusted prevalence maps for 2003, 2008 and 2014 KDHS obtained after adjusting for other potential confounders including socio-demographic variables, space, time and space-time interactions.

Figure 6 offers a clearer picture of the variations in FGM/C prevalence among 0–14 years old girls in Kenya having taken other confounding factors into account. In 2003, an estimated mean posterior prevalence of about 67% and 15% were found in the North-eastern and Nyanza regions, respectively. Also, in 2008, FGM/C prevalence remained high in North-eastern and Nyanza regions, and increased slightly in Eastern regions with a conspicuous reduction in Coast. There was an overall decline in prevalence in 2014 KDHS, although the rate for North-eastern region remained high at about 41%.

In Figure 8 we show how the prevalence of FGM/C has evolved temporally having taken other potential confounders into account. Also, we show the overall non-linear effects of mother’s age on her daughter’s likelihood of undergoing FGM/C across the survey years considered in the pooled dataset.

Figure 8 (left panel) showed that the likelihood of a girl being cut decreased steadily across the years and even more rapidly between 2008 and 2014. On the other hand, the right panel of Figure 7 shows that after accounting for other potential confounders in the pooled dataset, the likelihood of a girl being cut increased with her mother’s age.

## 4. Discussion

Our study used a spatial epidemiological approach to analyse the social norms influences on the prevalence of FGM/C among 0–14 years old girls in Kenya. Results show that FGM/C risk factors attributable to such influences were randomly distributed across the country’s regions. The use of Bayesian spatial modelling to account for this relationship is an important contribution to a robust evidence base, which highlights the importance of community-level influences on a wide range of outcomes including religion, observed in other countries (Kandala and Shell-Duncan 2019). More importantly, this methodological approach, rarely used in the ecological research on FGM/C, adds additional evidence to the more recent body of research testing the prediction of social norms theory in countries such as Senegal and the Gambia (Shell-Duncan et al. 2011).

The picture that emerged from our study is that normative influences are key drivers of FGM/C in Kenya. Having a circumcised mother, living in a community with a high proportion of cut women, or high proportion of women who supported FGM/C continuation, and high proportion of women who believed FGM/C was a religious obligation, increased the chances of a girl being cut.

Fully adjusted models further showed that being a Muslim, belonging to Kisii and Somali ethnic group and living in the North-eastern or Nyanza region made a girl more likely to be cut than her counterparts. In addition, living in urban area and belonging to a Middle wealth quintile household increased a girl’s risk of experiencing FGM/C.

Surprisingly, it was found that a girl whose mother had higher education, or no education had increased risk of being cut than her peers with a similar situation. This result indicates that FGM/C is independent of woman’s educational level with respect to the evidence from the 2014 KDHS. The results from the pooled data, however, showed that increase in mother’s level of education, decreased her daughter’s likelihood of being cut. This finding represents an overall effect of mother’s education from 2003 through to 2014 survey years.

We also found that the likelihood of a girl experiencing FGM/C decreased as the ethnic fractionalization index EFI increased. This finding supports the concept of assimilative change and shifting reference groups. Assimilative change (also called unprogrammed change by Mackie [42]), involves processes of cultural borrowing or innovation, along with reassessment of prior norms. Challenging the view that social norms are deeply rooted, and static or unchanging, scholarship has highlighted the fluidity of social norms, and the dynamic processes in which meanings can accrue, be altered, or discarded [43,44,45,46,47,48]. Recent studies on FGM/C have shown that far from being a static tradition rooted in values from the past, norms and meanings associated with the practice are subject to ongoing scrutiny and revision in light of shifting goals and social circumstances. Moreover, existing cultural heterogeneity may provide a valuable, though often overlooked, resource for programmed change efforts [39,40,49,50].

Assimilation change may occur more readily in communities with geographic proximity with non-practicing groups. Particularly where this results in social interactions between practicing and non-practicing families, people may have a greater opportunity to see that girls and women who are not cut do not experience negative sanctions such as ostracization. This may result in people questioning normative expectations and weakened support for the continuation of FGM/C. This hypothesis is consistent with the findings of Cloward, whose study in three Kenyan communities showed that higher interethnic interaction was associated with a more rapid shift away from norms in support of FGM/C [29]. Additionally, it is commonly posited that norm change may be more likely to occur among people exposed to international norms opposing FGM/C, particularly through exposure to messages in the media [30,31,32]. On the other hand, shifting reference groups supports change in social norms which involved shifting to a different reference group. Reference group members may include people from a variety of social circles: close friends and family, residents of one’s community, peers from school, work colleagues, and fellow members of a church or mosque. Since different social norms may be influenced by different reference groups, one possibility is for change to align with groups that tolerate a desired change. In regions where there is little ethnic diversity, and where the prevalence of FGM/C is near-universal, there may be little opportunity to shift to a reference group that does not uphold the practice. However, where there is ethnic mixture involving people who do and do not practice FGM/C, such a shift may be possible. More broadly, a lower local prevalence of FGM/C may indicate the possibility of realigning one’s reference group to minimize social sanctions. Finally, where migration is more common, there may be greater opportunities for families who oppose FGM/C to join a non-cutting community.

The maps of the unobserved spatial effects show clearly that highest likelihoods of daughter’s FGM/C are found mainly in North-Eastern Kenya. Notably, these are regions whose residents are primarily ethnic Somalis (North-eastern) and Kisii (Kisii and Nyamira Counties).

### Limitation

This study has two main limitations. The KDHS rely on self-reported data on FGM/C for women and mother-reported information on the cutting status of daughters and were not corroborated by physical examinations by trained health professionals. These reports need to be treated with caution as inaccuracies can arise due to factors such as recall bias about an event that occurred many years ago, courtesy bias, and unwillingness to disclose cutting status after criminalization of FGM/C [51]. A second limitation of the study is due to the different definitions of the outcome variable adopted in earlier surveys. Caution should, therefore, be applied while interpreting findings from this study.

## 5. Conclusions

This study confirms that normative influences measured by a woman’s FGM/C status, the proportion of cut women in the community, the proportion of women who supported FGM/C continuation and the proportion of women who believed that FGM/C was a religious obligation, were key drivers of FGM/C among 0–14 years old girls in Kenya. Prevalence varied across and within strata, although there was an overall decrease in National prevalence from 1998 to 2014. Higher risks of FGM/C were found among Kisii and Somali ethnic group and among Muslims. The study identified the North-eastern region as a hotspot geographical location where girls aged 0–14 years have the highest likelihood of experiencing FGM/C.

These findings offered clearer picture on the dynamics of FGM/C among 0–14 years old girls in Kenya and will aid targeted interventions through bespoke policymaking and implementations.

## Figures and Tables

**Figure 1 ijerph-16-04155-f001:**
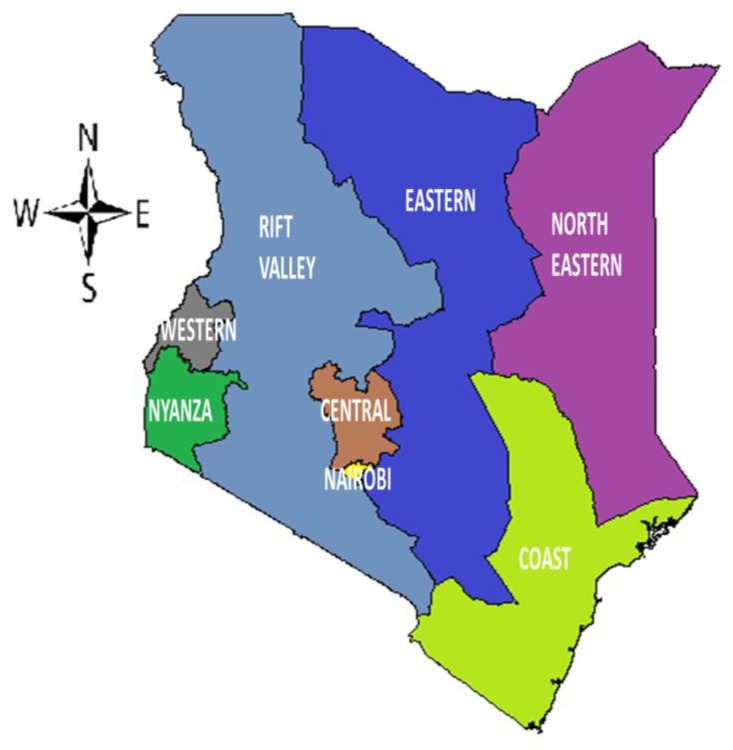
Map of Kenya showing the eight administrative regions.

**Figure 2 ijerph-16-04155-f002:**
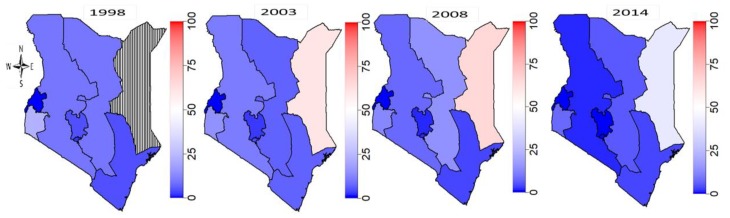
Evolution of FGM/C prevalence among 0–14 years old girls in Kenya from 1998 to 2014. Note that in 1998, there were no surveys in the North Eastern region hence the black-white stripes. Across the years, red regions had highest prevalence which decreased in magnitude as it fades through red to deep blue.

**Figure 3 ijerph-16-04155-f003:**
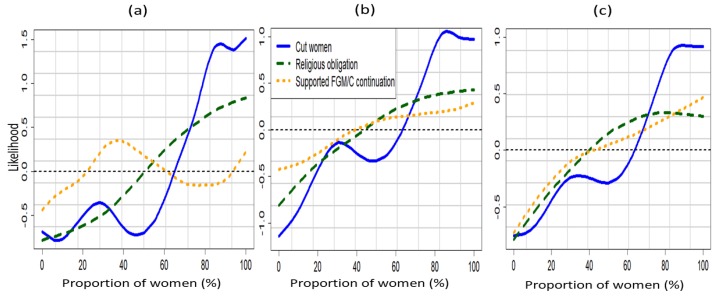
Non-linear effects of the proportions of women in a girl’s community who were cut (blue), who supported the continuation of FGM/C (orange), and who believed that FGM/C was a religious obligation for (**a**) Model A, (**b**) Model B and (**c**) Model C. Evidence from the 2014 KDHS.

**Figure 4 ijerph-16-04155-f004:**
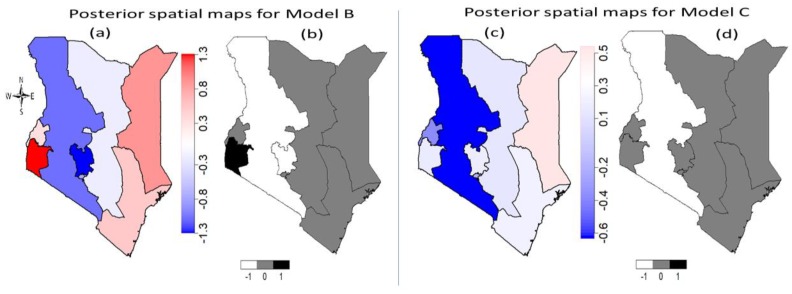
Posterior risk maps [(**a**) and (**c**)] of Kenyan 0–14 years old girls’ FGM/C with the corresponding 95% (right) [(**b**) and (**d**)] posterior significance maps for Model B (left panel) and Model C (right panel). Deep blue to red corresponds to low risk to high risk. Black colour indicates significantly high-risk regions, white colour indicates significantly low risk regions and grey colour indicates nonsignificant regions.

**Figure 5 ijerph-16-04155-f005:**
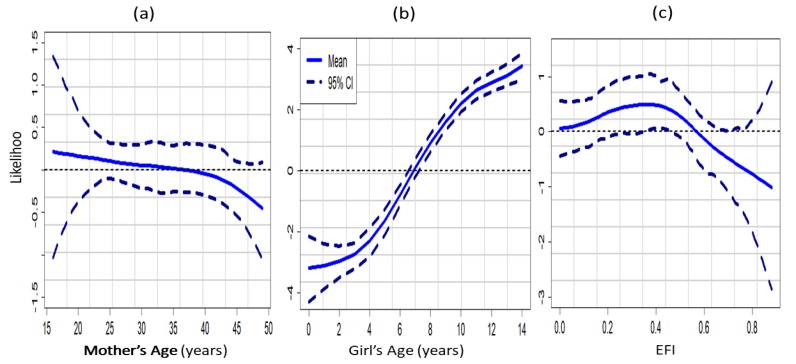
Non-linear effects on a girl’s likelihood of experiencing FGM/C of mother’s age (**a**); girl’s age (**b**) and ethnic fractionalization index (EFI) (**c**). Evidence from the 2014 KDHS Model C.

**Figure 6 ijerph-16-04155-f006:**
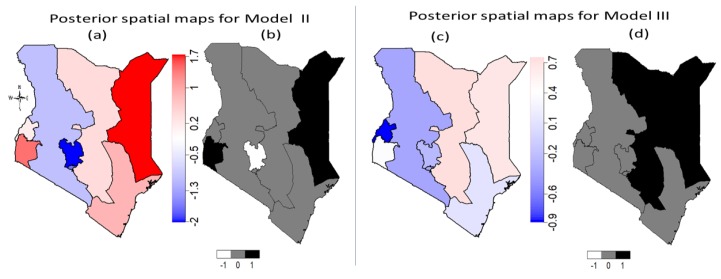
Posterior risk maps ((**a**) and (**c**)) of Kenyan 0–14 years old girls’ FGM/C with the corresponding 95% (right (**b**) and (**d**)) posterior significance maps for Model II (left panel) and Model III (right panel). Deep blue to red corresponds to low risk to high risk. Black colour in (**b**) and (**d**) indicates significantly high-risk regions, white colour indicates significantly low risk regions and grey colour indicates nonsignificant regions. Evidence from the pooled 2003 to 2014 KDHS.

**Figure 7 ijerph-16-04155-f007:**
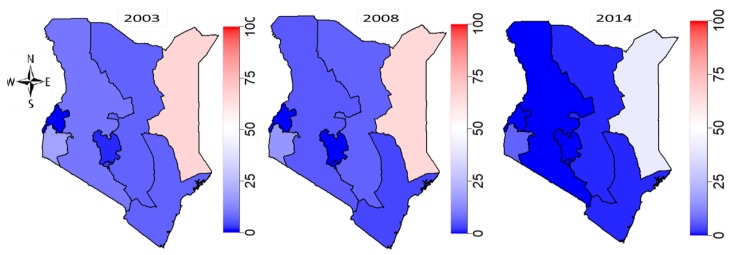
Predicted fully adjusted FGM/C prevalence among 0–14 years old girls in Kenya from 2003 to 2014 from the best fit model (Model III) of the pooled 2003 to 2014 data. Across the years, red regions had highest prevalence, while deep blue regions had lowest prevalence.

**Figure 8 ijerph-16-04155-f008:**
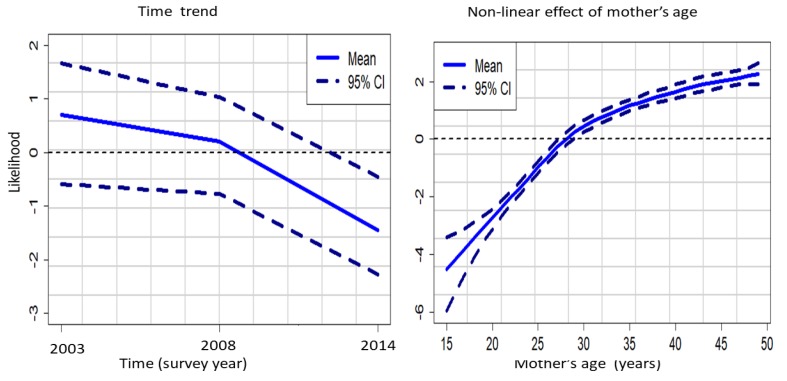
Time trend (left panel) and non-linear effect of mother’s age (right panel). Evidence from the KDHS 2003 to 2014 pooled data Model C*.

**Table 1 ijerph-16-04155-t001:** National prevalence and girl’s female genital mutilation/cutting (FGM/C) prevalence by normative influence variables across the four selected Demographic and Health Surveys (DHS) surveys in Kenya, Kenya Demographic and Health Surveys (KDHS) 1998 to 2014.

Factor	Level	1998 KDHS (n = 4069, FGM/C = 9.9%)	2003 KDHS (n = 4048, FGM/C = 9.4%)	2008 KDHS (n = 7195, FGM/C = 7.6%)	2014 KDHS (n = 12,434, FGM/C = 3.0%)
**Mother’s FGM/C status**	*Cut*	22.0	23.5	21.5	9.9
	*Uncut*	0.3	0.4	0.1	0.2
**Mother’s support for FGM/C continuation**	*Continued*	29.2	--	41.8	23.1
	*Undecided*	5.1	--	4.0	1.1
**Mother’s belief that FGM/C is required by religion**	*Required*	--	--	46.7	27.8
	*Not required*	--	--	4.5	1.4

Note: The percentages are not complementary and thus did not add up to 100% since what is shown is the percentage of women in the given category whose daughters were cut.

**Table 2 ijerph-16-04155-t002:** Deviance information criterion (DIC) and effective sample size (pD) from the three models fitted on the 2014 KDHS data and the combined 2003 to 2014 data.

Data	Model	Description	DIC	Effective Size (PD)
**2014** **KDHS**	**Model A**	Normative influence variables only	3551.4	19.7
	**Model B**	Normative influence variables and total space	1805.3	62.6
	**Model C**	Normative influence variables, space and other individual-level covariates	1718.6	117.1
**2003 to 2014 Pooled Data**	**Model I**	Normative influence variables only.	8104.9	8.0
	**Model II**	Normative influence variables, space without time and space-time interaction.	8108.1	9.7
	**Model III**	Normative influence variables, space and time, space-time interaction and other individual-level covariates	6501.4	44.42

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
