# Peer review of "A Spatial Analysis of the Prevalence of Female Genital Mutilation/Cutting among 0–14-Year-Old Girls in Kenya"

_ijerph, 2019, doi:10.3390/ijerph16214155_

Round 1
Reviewer 1 Report
The authors have reviewed the social scenario of female genital mutilation in the sub-saharan continent, and the effects on future genital mutilation of various factors including female education, whether they have been circumcised, on future genital mutilation. The review is concise and throws light on aspects that need to be addressed to control this practice.
The bibliography needs to be corrected (serial number is repeated in most references)

Author Response
3. Author's Reply to the Review Report (Reviewer 1)
The bibliography needs to be corrected (serial number is repeated in most references)
RW1.1: The authors have reviewed the social scenario of female genital mutilation in the sub-saharan continent, and the effects on future genital mutilation of various factors including female education, whether they have been circumcised, on future genital mutilation. The review is concise and throws light on aspects that need to be addressed to control this practice.
REPLY: The authors highly appreciate this. Thank you.
RW1.2: The bibliography needs to be corrected (serial number is repeated in most references)
REPLY: Thank you. This has been revisited and reflected in the main manuscript.

Reviewer 2 Report
Thank you for writing on this very important issue. Female Genital Mutilation has always been a social and health concern wherever communities practicing this culture are found.
However, I do have some concerns with the usage of so many statistical models which may discourage readers with limited statistical knowledge to continue reading the manuscript. The title of your paper may attract readers, especially activists against FGM but reading through the paper, understanding and using information in the manuscript may not be simple for this category of people.
Reviewer 3 Report
This paper discusses an important public health issue and was interesting to read. The authors analyzed the impact of individual level and community level factors on female genital mutilation, whilst accounting for potential spatial and temporal variations. The analysis and presentation of findings could be much improved if the authors address the concerns stated below.
Major comments
The model presented in equation 2 has some deficiencies. First, there is no description of how the spatiotemporal term was modelled. Second, the indexing of the variables is conflated. The prevalence p_i, for example, should be indexed as p_ist, where i refers to ith child, s - the area the ith child comes from and t - the DHS year for the data point. Some terms on the RHS of the equation needs to be relabelled accordingly. The authors used a BYM specification to model spatial variation. However, better choices are possible (see Lee, D . A comparison of conditional autoregressive models used in Bayesian disease mapping. Spat Spatiotemporal Epidemiol 2011; 2: 79–89.) The authors should state the reasons behind the choice made. The spatial analysis was carried out at admin level one which is usually of less interest (large admin areas concealing a lot of heterogeneities) since estimates can be directly produced at this level from survey data. One would expect that the analysis would produce smoother maps in comparison with the raw data presented in Figure 2 but this was not the case as Figure 7 shows. If the spatial analysis is one of the main goals of the study, this needs to be addressed. In the interpretations presented at page 9 and the subsequent pages, the authors do not distinguish between associations that were significant and those that were not. For example, between lines 328 amd 333, the authors reported a POR of 3.94 (95% CI = 2.05, 7.63) for model A and 1.97 (95% CI = 0.69, 6.01) for model C (?) and went ahead to assert that the likelihood of cutting a girl whose mother was cut was high in both cases; when clearly, the likelihood of being cut was not significant in the latter. Many other interpretations followed the same pattern. The authors need to address this by laying more emphasis on the associations that were significant. In addition, I do not understand the value of detailed interpretation of models that yielded poorer fit of the data. In lines 346-348, the claim "significant increase in likelihood of being cut" is not immediately obvious when examining Figure 3 as the y-axes are not shown on a probability or more interpretable scale. The same thing also applies to Figure 8. In lines 356 and 357, the expression "95% posterior probability maps" is odd. It is not clear what this means. The range of values plotted (Figure 4) does not indicate that these are probabilities. In panels (a) and (c), it is not clear whether the values plotted represent both the unstructured and structured spatial terms or only the latter. These should be clarified. The authors could consider including the code used in the analysis in the submission to help the reader understand the details of the analyses carried out.
Minor comments
Line 63 ...study by Achia - reference not provided Figures 2 and 7, use the same scale for comparability Line 459 - no figure 7a In the abstract, it was stated that the analyses included DHS data from 1998-2014. I think that this is misleading as 1998 data was only used for descriptive analysis and not for modelling.
